# Development of multiplex real-time PCR for rapid identification and quantitative analysis of *Aspergillus* species

Won-Bok Kim[1], Chulmin Park[1], Sung-Yeon Cho[1,2,3], Hye-Sun Chun[1], Dong-Gun Lee[1,2,3]*

1 Vaccine Bio Research Institute, College of Medicine, The Catholic University of Korea, Seoul, Korea,
2 Division of Infectious Diseases, Department of Internal Medicine, College of Medicine, The Catholic University of Korea, Seoul, Korea, 3 Catholic Hematology Hospital, Seoul St. Mary's Hospital, College of Medicine, The Catholic University of Korea, Seoul, Korea

* symonlee@catholic.ac.kr

**Data Availability Statement:** All relevant data are within the paper and its Supporting Information files.

## Abstract

The identification of *Aspergillus* species and azole resistance is highly important for the treatment of invasive aspergillosis (IA), which requires improvements in current fungal diagnostic methods. We aimed to develop multiplex real-time PCR to identify major *Aspergillus* section and azole resistance. *BenA* and *cyp51A* genes were used to design primers, probes, and control DNA for multiplex PCR. Qualitative and quantitative analysis was conducted for 71 *Aspergillus* and 47 non-*Aspergillus* isolates. Further, the limit of detection (LOD) and limit of quantitation (LOQ) from hyphae or conidia were determined according to the culture time. Newly developed real-time PCR showed 100% specificity to each *Aspergillus* section (*Fumigati*, *Nigri*, *Flavi*, and *Terrei*), without cross-reaction between different sections. In quantitative analysis of sensitivity measurements, LOD and LOQ were 40 fg and 400 fg, respectively. Melting temperature analysis of the *cyp51A* promoter to identify azole resistance showed temperatures of 83.0 ± 0.3°C and 85.6 ± 0.6°C for susceptible *A. fumigatus* and resistant isolates with TR34 mutation, respectively. The minimum culture time and fungal colony size required for successful detection were 24 h and 0.4 cm in diameter, respectively. The developed multiplex real-time PCR can identify common *Aspergillus* sections quantitatively and detect presence of the TR34 mutation. Further, this method shows high sensitivity and specificity, allowing successful detection of early-stage fungal colonies within a day of incubation. These results can provide a template for rapid and accurate diagnosis of IA.

## Introduction

Invasive aspergillosis (IA) is a fatal disease caused by *Aspergillus* species that occurs mainly in immunocompromised patients [1]. Common species that cause IA include *Aspergillus fumigatus*, *A. flavus*, *A. terreus*, and *A. niger*. Recently, global studies have reported "cryptic species" such as *A. lentulus*, *A. udagawae*, and *A. tubingensis* and increasing antifungal resistance.

**Funding:** This research was supported by an industry-university research grant (5-2019-D0166-00004) through the Yuhan Corporation. The funder provided support in the form of salaries for authors [WBK, CP and HSC], but did not have any additional role in the study design, data collection, and analysis, decision to publish, or preparation of the manuscript. The specific roles of these authors are articulated in the 'Author contributions' section.

**Competing interests:** DGL has served as a consultant for Astellas, MSD, Pfizer, Gilead, and Yuhan; has served as a board member for Astellas, MSD, Gilead and Yuhan; and has received research support, travel support and payment for lectures, including service on Speaker's bureaus, from Astellas, MSD, Pfizer, Gilead, and Yuhan, outside the submitted work. SYC has received research support and payment for lectures from MSD, Gilead, and Astellas, outside the submitted work. DGL, WBK, CP, and SYC have a pending patent application on the development of multiplex real-time PCR in Korea (Korean patent pending no. 10-2019-0119028). However, the funder did not any role in the pending patent. This does not alter our adherence to PLOS ONE policies on sharing data and materials. The authors declare that they have no further competing interests.

Above all, rapid and accurate fungal diagnosis is important, as the morbidity and mortality of IA remain high, and diagnosis and treatment impart a significant economic burden [2–5].

The method of *Aspergillus* species identification includes morphologic identification of fungi through culture and molecular identification via the polymerase chain reaction (PCR) [5, 6]. The former method is currently the gold standard; however, its use can depend largely on clinical specimen quality and the proficiency of the microbiology test personnel [6, 7] The latter method, molecular identification of filamentous ascomycetes, is mainly conducted through sequence analysis of the internal transcribed spacer (*ITS*) region, and *β-tubulin* (*benA*) and *calmodulin* (*CaM*) genes are also used for cryptic species-level identification [2, 8, 9]. However sequencing-based identification is time-consuming, as it takes 5–17 days to reach the final diagnosis; specifically, it takes 3–14 days to produce conidia following fungal culture from the clinical sample, followed by an additional 2–3 days to extract DNA from the conidia and obtain sequencing analysis results [10, 11].

Therefore, to take advantage and overcome the shortcomings of the multi-step molecular diagnosis method, there have been many efforts to explore new diagnostic approaches [6–14]. In this study, we aimed to develop a method for rapid molecular identification by using a multiplex platform that allows for quantitative analysis of major *Aspergillus* sections and rapid detection of azole resistance.

## Material & method

### Fungal isolates and culture

The fungal isolates used in this study were representative strains of clinical and environmental isolates stored anonymously, and standard strains including *A. fumigatus* (ATCC 16424; American Type Culture Collection, Manassas, VA, USA), *A. terreus* (ATCC 10690), *A. flavus* (ATCC 16883), and *A. niger* (ATCC 16888) [2, 15]. The representative *Aspergillus* isolates were selected for each sequence type according to *benA* and *cyp51A* sequencing results. The *Aspergillus* isolates used in this study included 20 strains of *A. fumigatus*, 2 strains of *A. lentulus*, 1 strain of *A. turcosus*, 1 strain of *A. udagawae*, 2 strains of *A. luchuensis*, 11 strains of *A. awamori*, 11 strains of *A. niger*, 12 strains of *A. tubingensis*, 4 strains of *A. flavus*, 1 strain of *A. nidulans*, 2 strains of *A. sydowii*, 3 strains of *A. terreus*, and 1 strain of *A. subramanianii*, which were registered to GenBank in our previous study [2]. In addition, 38 non-*Aspergillus* filamentous ascomycetes (No. 1–38), 1 non-filamentous ascomycetes (No. 39), and 8 non-ascomycetes molds (No. 40–47) were used to measure the specificity of the developed molecular identification method (S1 Table). Fungal isolates were cultured using Sabouraud's dextrose agar medium (Becton-Dickinson Labware, Franklin Lakes, NJ, USA) in an incubator at 35°C for 1–10 days. Conidia or hyphae were harvested for genomic DNA (gDNA) extraction using 0.85% NaCl with 0.05% Tween 20. Pellets of conidia or hyphae were stored at −80°C until DNA extraction. The Institutional Review Board of Seoul St. Mary's Hospital approved the research protocol of this study (KC16SISI0307) and waived informed consent for use of fungal isolates stored anonymized.

### DNA extraction and purification

To optimize DNA extraction from fungal isolates, the following steps were conducted. In brief, the hyphae pellets stored in the deep freezer were placed in 500 μL Trizol reagent (Invitrogen, Carlsbad, CA, USA), and the conidia pellets were placed in 500 μL of yeast cell lysis solution (MasterPure™ Yeast DNA Purification Kit, Epicentre, Madison, WI, USA); subsequently, the pellets were subjected to two freeze-thaw cycles. After adding RNase A (Epicentre), incubating at 5 min in room temperature. And bead-beating (1.0 mm zirconium beads; Sigma-Aldrich,

St. Louis, MO, USA) was repeated 3–5 times at 12,500 rpm for 5 min and cooled on ice at 1 min in between time. Then, the supernatant was obtained after centrifugation ($>$12,000 $\times$ $g$, 4°C, 5 min). In the method using Trizol reagent, chloroform (Sigma-Aldrich) was added and reacted at room temperature for 5 min; in the method using yeast cell lysis solution, MPC protein precipitation reagent (Epicentre) was added and vortexed. The sample was centrifuged ($>$12,000 $\times$ $g$, 4°C, 15 min) to collect the supernatant, which was then treated with isopropanol (Sigma-Aldrich). A QIAamp DNA Mini spin column (Qiagen, Hildren, Germany) was used to purify fungal DNA. The extracted DNA concentration was measured using Quant-iT™ Pico-Green™ dsDNA Reagent (Invitrogen) according to the manufacturer's protocol. In the case that further DNA purification was required, the DNeasy PowerClean Pro CleanUp Kit (Qiagen) was used according to the manufacturer's protocol. The extracted DNA was stored at −20°C before the next experiment.

## Design of primers and probes

In this study, the sequences of clinical and environmental *Aspergillus* isolates were analyzed to design new primers for amplifying the relatively short sequence of exons 5–6 of the *benA* gene (for *A. fumigatus* AF293 strain *benA* gene, GCA_000002655.1_ASM265v1, locus_tag = AFUA_1G10910, location no. 544–803) (Table 1). The nucleotide sequence of *Aspergillus* isolates was analyzed using MegAlign Pro 15 (DNASTAR, Madison, WI, USA). A probe that could detect filamentous ascomycetes was designed in exon 5, and other probes were designed for the intron between exons 5 and 6 (S1 Fig). For the amplification and simultaneous detection of the *benA* target, dimer and hairpin formation of primers and probes was measured, and heterodimer delta G values ranged from −3.6 to −8.9 kcal/mol of five probe.

We designed two staged probe sets: For the first stage, primer and probe sequences specific to four major *Aspergillus* sections (*Fumigati*, *Nigri*, *Terrei*, and *Flavi*) were designed, and a control probe sequence was also determined to identify filamentous ascomycetes. Based on the analysis using the OligoAnalyzer web program (https://sg.idtdna.com/pages/tools/oligoanalyzer) for each primer and probe, the final biomarkers were developed for multiplex

**Table 1. Nucleic acid sequences of biomarkers (primers or/and probes set) designed for multiplex identification in this study.**

| PCR | Target species | Primers / probes | Sequence (5′ → 3′) | Target loci | Product size | Reference |
|---|---|---|---|---|---|---|
| Multiplex real-time PCR primer | *Aspergillus spp.* | benA F3 | `TCGGTGTAGTGACCCTTGG` | *β-tubulin (benA)* | 254 ~ 272 bp | In this study |
| | | benA R2 | `GCTGGAGCGYATGAACGTCT` | | | |
| | *A. tubingensis* | Tubcyp 1F46 | `CTCGTTGCGATAGTCTTKAATGT` | *cyp51A* | 263 bp | |
| | | Tubcyp 1R308 | `GCCCAAGTATACGGTKGTCTTTT` | | | |
| | *A. fumigatus* | TR F1 | `TAATCGCAGCACCACTTCAG` | | WT[‡]: 111 bp | [16] |
| | | TR R1 | `AGGGTGTATGGTATGCTGGAA` | | TR34 : 145 bp | In this study |
| Hydrolysis probe | Ascomycetes | Asco 1F9 | `6-FAM-AVACGAAGTTGTCGGGRC-TAMRA` | *β-tubulin (benA)* | 18 bp | In this study |
| | Section *Fumigati* | Fumi 1R2 | `HEX-CGGCAACATCTCACGATCTGACTCGC-BHQ1` | | 26 bp | |
| | Section *Nigri* | Nig 1R26 | `6-FAM-ACTTCAGCAGGCTAGCGGTAACAAGT-TAMRA` | | 26 bp | |
| | Section *Flavi* | Flavi 1F18 | `HEX-CGGTCAGGAGTTGCAAAGCGTTTTCA-BHQ1` | | 26 bp | |
| | Section *Terrei* | Terrei 1R29 | `6-FAM-ACCATCCTGGGACAGATTCTYCACGC-TAMRA` | | 26 bp | |
| | *A. tubingensis* | Tubcyp 201 | `6-FAM-GTCAGYCACTGTCCATATGCGATTG-TAMRA`[†] | *cyp51A* | 25 bp | |

[†] The underlined area is a probe synthesized by LNA.

[‡] Wild type

molecular identification (Table 1). The specificity of the synthesized probe was evaluated using BLASTn (https://blast.ncbi.nlm.nih.gov/Blast.cgi), and the presence of cross-reactions between sections was identified. For the second stage, primers and probes specific for both TR mutations in the *cyp51A* promoter and *A. tubingensis* were designed (Table 1). The *A. tubingensis* probe was designed for a specific sequence in the *cyp51A* gene using locked nucleic acid to increase specificity. Promoter mutation was detected by melting temperature analysis instead of the probe detection method.

## Synthesized the control DNA

More controls need more wells. In clinical settings, more control leads to fewer test samples, especially in quantitative analysis. Therefore, we designed a synthesized control including all probes (5 probes). At first, we aligned each consensus sequence of *benA* in *Aspergillus* species (S1 Fig). We used *benA* sequence of *A. fumigatus* as the template and designed to have each probe sequence (conserved probe of filamentous ascomycetes, *Fumigati*, *Nigri*, *Flavi*, and *Terrei*) (Table 2). And then, the designed DNA was synthesized genetically (Bionics Corporation, Seoul). We cloned it into the pUC57 plasmid, and it was transformed into *Escherichia coli* (*E. coli*) BL21 strain. Plasmid DNA of transformants was extracted, tested for the multiplex, and used as control DNA.

## Qualitative and quantitative analysis by multiplex real-time PCR

Qualitative and quantitative analyses were conducted using the LightCycler® 480 Probes Master Kit (Roche, Basel, Switzerland) and Roche LightCycler® 480 Instrument II (Roche). The primer and probe concentrations used in the experiment were 0.5 μM and 0.1 μM, respectively. The amplification process was conducted as follows: 10 min of pre-denaturation at 95°C followed by 40 cycles of denaturation at 95°C for 25 s, annealing at 58°C for 30 s, and extension at 72°C for 35 s. For quantitative analysis, 4 ng fungal gDNA and 4 pg *Aspergillus* positive control DNA were each subjected to 10-fold serial dilutions to reach 4 fg/μL and 4 ag/μL, respectively. Melting peak analysis was conducted using the LightCycler® 480 SYBR Green I Master Kit (Roche) and Roche LightCycler® 480 Instrument II (Roche).

The results were analyzed using LightCycler® 480 Software version 1.5.0 SP3 (Roche). The quantification cycle (Cq) was calculated automatically by the program based on curve fitting to the baseline curve subtraction. A standard curve was obtained using the measured results, and the slope value, mean Cq value (MCq), and standard deviation (SD) were calculated. The amplification efficiency (Efficiency, E [%]) was calculated by the following equation: $[10^{(-1/\text{slope})} - 1] \times 100$. Sensitivity analysis of the developed identification method was conducted using the limit of detection (LOD) and limit of quantitation (LOQ). The LOD was set to the minimum value detected above 95% of analysis results (<5% false negative result). The LOQ was determined to be a value of MCq − (2 × SD) below 35 [17, 18].

**Table 2. Positive control nucleic acid sequences for *Aspergillus* real-time PCR assays.**

| Oligo | DNA Sequence (5′ → 3′) |
|---|---|
| *Aspergillus* Positive Control DNA (276 bp) | ATCGTAAGCTTT<u>TCGGTGTAGTGACCCTTGG</u>CCCAGCCGA<u>AGACGAAGTTGTCGGGACGGACGGT</u>CAGGAGTTGCAAAGCG |
| | benA F3 primer Asco 1F9 probe Flavi 1F18 probe |
| | TTTTCAGCACGGACAACTCAGGGACGGTGTGATCTAACCTCATGGTAC<u>GCGTGGAGAATCTGTCCCAGGATGGT</u>AGAACGG |
| | Terrei 1R29 probe |
| | CACGAGGTA<u>GCGAGTCAGATCGTGAGATGTTGCC</u>GAAATAGTATAAATCAAGAG<u>ACTTGTTACCGCTAGCCTGCTGAAGTAA</u> |
| | Fumi 1R2 probe Nig 1R26 probe |
| | ATA<u>GACGTTCATGCGCTCCAGC</u>GAGCTCATCGT |
| | benA R2 primer |

All experiments were conducted in accordance with the "Minimum Information for the Publication of Real-Time Quantitative PCR Experiments" (MIQE) guidelines [19, 20]. In addition, the experiment was conducted in a biosafety cabinet in an area separate from the DNA extraction area to prevent contamination from aerosol and carryover. To remove contaminants on the surface, 1.5% sodium hypochlorite (NaOCl) was used, followed by 70% EtOH. The experiment was conducted using filter tips (TipOne; StarLab, Hamburg, Germany) after removing all possible contaminants.

## Inhibition testing

The SPUD assay was used as an exogenous amplification control [19]. The SPUD inhibition assay results generate the expected baseline Cq value for the SPUD amplicon and use it as the basis for the uninhibited assay. Based on this, Cq values exceeding 1 cycle or more of other DNA amplifications indicate the presence of a PCR inhibitor. The reaction of each qPCR was carried out using the LightCycler 480 SYBR Green I Master Kit (Roche). The SPUD plasmid DNA was $1.3 \times 10^5$ genome copy number (copy/μL) and the primer 0.5 μM. And sample extract was added 4ng gDNA. Thermal cycling conditions were 10 min of pre-denaturation at 95˚C followed by 40 cycles of denaturation at 95˚C for 10 s, annealing at 60˚C for 10 s, and extension at 72˚C for 20 s.

## Results

### Qualitative analysis to determine probe specificity

Multiplex real-time PCR of 118 fungal DNA samples showed that the probes specifically amplified the *Aspergillus* section as presented in Table 3. No amplification was observed in non-*Aspergillus* isolates. 1 non-filamentous ascomycetes (No. 39), and 8 non-ascomycetes molds (No. 40~47) showed no amplification of all primers and probes, including the Asco 1F9 probe. Further, the *Aspergillus* positive control DNA was amplified in all primers and probes designed for the first stage, and there was no interference between probes. For the second stage, real-time PCR was performed for 20 *A. fumigatus* (17 wild-type and 3 mutant strains), 2 *A. lentulus*, 1 *A. turcosus*, and 1 *A. udagawae* strains, and the melting peaks were analyzed. The melting temperature was 83.0 ± 0.3˚C in *A. fumigatus* wild-type samples and 85.6 ± 0.6˚C in mutant isolates with TR34 (S2 Fig). *Aspergillus* section *Fumigati* isolates were not amplified except for *A. fumigatus*. In addition, one primer pair (Tubcyp 1F46 and Tubcyp 1R308) and one probe (Tubcyp 201) designed to identify *A. tubingensis* from *Aspergillus* section *Nigri* were tested on 36 *Aspergillus* isolates of section *Nigri* in real-time PCR analysis. All 12 *A. tubingensis*

**Table 3. Specific amplification of the designed probes.**

| Probe (n) | *Aspergillus* section | | | | | Non-*Aspergillus* (47) |
|---|---|---|---|---|---|---|
| | *Fumigati* (24) | *Nigri* (36) | *Flavi* (4) | *Terrei* (3) | other (4) | |
| Asco 1F9 | 24 / 24 (100%) | 36 / 36 (100%) | 4 / 4 (100%) | 3 / 3 (100%) | 4 / 4 (100%) | 30 / 38[†] (63.82%) |
| Fumi 1R2 | 24 / 24 (100%) | 0 / 36 (0%) | 0 / 4 (0%) | 0 / 3 (0%) | 0 / 4 (0%) | 0 / 47 (0%) |
| Nig 1R26 | 0 / 24 (0%) | 36 / 36 (100%) | 0 / 4 (0%) | 0 / 3 (0%) | 0 / 4 (0%) | 0 / 47 (0%) |
| Flavi 1F18 | 0 / 24 (0%) | 0 / 36 (0%) | 4 / 4 (100%) | 0 / 3 (0%) | 0 / 4 (0%) | 0 / 47 (0%) |
| Ter 1R29 | 0 / 24 (0%) | 0 / 36 (0%) | 0 / 4 (0%) | 3 / 3 (100%) | 0 / 4 (0%) | 0 / 47 (0%) |
| Tubcyp 201 | 0 / 24 (0%) | 12 / 12 (100%) | 0 / 4 (0%) | 0 / 3 (0%) | 0 / 4 (0%) | 0 / 47 (0%) |

[†] Ratio of non-*Aspergillus* Filamentous ascomycetes

isolates with different *cyp51A* sequence types were amplified (Table 3); other isolates were not amplified.

## Probe sensitivity and efficiency measurement

Quantitative analysis results of gDNA of major *Aspergillus* sections are shown in Fig 1. There was no interference during the amplification of two probes analyzed in the multiplex assay, and gDNA was amplified from 4 ng/μL to 40 fg/μL in all probes. Similarly, amplification occurred without interference in the positive control DNA, which was amplified from 4 pg/μL to 40 ag/μL in all probes. The standard curve obtained from plotting gDNA concentration on the X-axis and Cq value on the Y-axis yielded slope values, intercept values, and $R^2$ –in the ranges of −3.2119 to −3.7230, 40.316 to 43.702, and 0.9979 to 0.9998, respectively. The amplification efficiency of each probe was found to be 89.8% ~ 104.8% in gDNA and 90.0% ~ 99.4% in Control DNA (Table 4). The LOD and LOQ of gDNA in all probes was 40 fg/μL (1.0–1.2 × $10^0$ genome copies/μL) and 400 fg/μL (1.0–1.2 × $10^1$ genome copies/μL), respectively. For positive control DNA, all probes showed similar correlations and regression analysis results, as shown in Table 5. According to the sensitivity analysis results using this data, the LOD and LOQ were calculated as 40 ag/μL (1.2 × $10^1$ genome copies/μL) and 400 ag/μL (1.2 × $10^2$ genome copies/μL), respectively (Table 5). The quantitative analysis results for *Aspergillus* gDNA and positive control DNA showed similar correlations and regression coefficients for all probes.

## Sensitivity measurement for molecular identification method by culture time

Four reference strains were used to measure the fungal colony diameter over the culture time. The size of *A. fumigatus*, *A. niger*, *A. terreus*, and *A. flavus* colonies was 0.2–1.5 cm at 18–24 h after incubation, 1.4–5.2 cm after 48 h, and 4.0–7.8 cm after 72 h. Further, conidia formation was observed after 72 h. Multiplex real-time PCR using DNA extracted from early-stage colonies showed successful amplification in hyphae-state after culturing for 24 h. The minimum colony diameter to confirm colony DNA amplification in all probes was 0.4 cm (confidence interval, CI > 95%), and the minimum culture time was 18–24 h. The result of multiplex real-

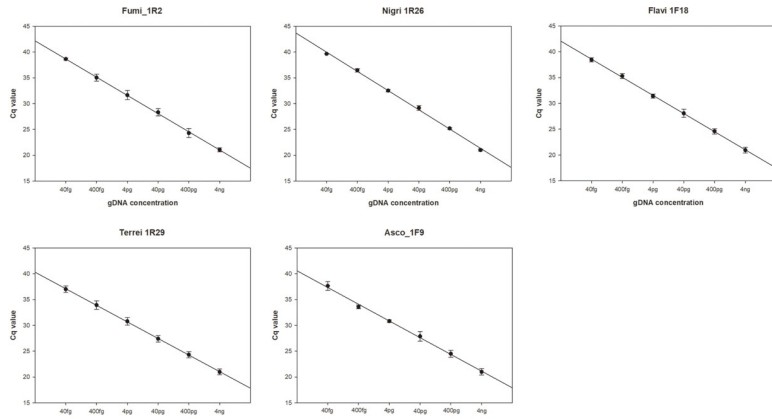

**Fig 1. Sensitivity measurements of *Aspergillus* gDNA and quantification curve.** *Aspergillus* gDNA and *Aspergillus* positive-control DNA were subjected to multiplex real-time PCR using Fumi 1R2, Nigri 1R26, Flavi 1F18, Terrei 1R29, Asco 1F9 and the results were presented as a standard curve. The spot is the average of Cq ± SD and n = 6. Abbreviations. gDNA; genomic DNA, SD; Standard Deviation.

**Table 4. Correlation and regression analysis between two DNA.**

| Probe name | Genomic DNA | | | | Control DNA | | | |
|---|---|---|---|---|---|---|---|---|
| | Range | E (%)[†] | Slope | R² | Range | E (%)[†] | Slope | R² |
| Fumi_1R2 | 21.1 ~ 38.6 (± 0.39) | 92.2 | -3.5245 | 0.9993 | 21.5 ~ 38.6 (± 0.20) | 90.1 | -3.5844 | 0.9998 |
| Nig_1R26 | 21.0 ~ 39.6 (± 0.20) | 89.8 | -3.5930 | 0.9983 | 21.1 ~ 39.1 (± 0.31) | 90.0 | -3.5866 | 0.997 |
| Flavi_1F18 | 20.9 ~ 37.4 (± 0.55) | 92.6 | -3.5132 | 0.9995 | 21.9 ~ 39.1 (± 0.74) | 94.0 | -3.4755 | 0.999 |
| Ter_1R29 | 20.9 ~ 37.0 (± 0.62) | 104.8 | -3.2119 | 0.9998 | 21.1 ~ 38.9 (± 0.73) | 92.5 | -3.5168 | 0.9978 |
| Asco_1F9 | 21.0 ~ 37.6 (± 0.87) | 103.5 | -3.2412 | 0.9979 | 22.0 ~ 38.5 (± 0.13) | 99.4 | -3.3373 | 0.9975 |

[†]E: Efficiency

time PCR of DNA from hyphae-state colonies of 0.4 cm in diameter for all isolates showed Cq values of 31.425 ± 0.245 for *A. fumigatus*, 32.345 ± 0.185 for *A. niger*, 32.63 ± 0.36 for *A. terreus*, and 32.16 ± 0.12 for *A. flavus*. This suggested that amplification occurred within the range of the standard curve of *Aspergillus* positive control DNA used as standard DNA, which is between approximately $1.0–1.2 \times 10^2$ and $1.0–1.2 \times 10^1$ copies/µL when calculated with respect to genomic DNA copy number.

## Inhibition

Each of the DNA extracts material was included in a SPUD assay and results indicated that no inhibition was present in gDNA samples (S3 Fig).

## Discussion

In this study, a molecular identification method for major *Aspergillus* sections (*Fumigati*, *Nigri*, *Flavi*, and *Terrei*) and filamentous ascomycetes were developed based on the *benA* gene, which reduced diagnosis time compared with the current methods of morphological or molecular identification based on ITS sequencing. Also, probe-based real time PCR can more sensitive, specific, fast, and used in a quantitative than multiplex PCR. For this reason, probe-based real time PCR was used in the design. In addition, TR mutation and the *cyp51A* gene were used to identify azole-resistant *A. fumigatus* and *A. tubingensis*. Instead of probe-based sequencing, melting temperature analysis methods were selected for faster and simultaneous analysis. The designed probes and primers were used for qualitative and quantitative analyses of 118 clinical and environmental isolates. Sensitivity and specificity were evaluated for the developed multiplex real-time PCR method. Further, both LOD and LOQ with minimum culture time were analyzed to evaluate to the applicability of the method for early diagnosis.

For the molecular identification of mold, there is a need to improve the process and purity of DNA extraction. Extracted DNA quality and impurity contents, such as PCR inhibitors, differ between each clinical and environmental isolate; thus, it is difficult to conduct quantitative

**Table 5. Correlation and regression analysis between two DNA.**

| gDNA / control DNA | Fumi 1R2 | Nig 1R26 | Flavi 1F18 | Ter 1R29 | Asco 1F9 |
|---|---|---|---|---|---|
| 4 ng / 4 pg | 21.07 / 21.50 | 21.01 / 20.12 | 20.94 / 21.92 | 20.98 / 21.20 | 20.93 / 22.08 |
| 400 pg / 400 fg | 24.30 / 25.21 | 25.21 / 24.41 | 24.59 / 25.00 | 24.31 / 24.41 | 24.50 / 25.45 |
| 40 pg / 40 fg | 28.34 / 29.01 | 29.17 / 28.70 | 28.07 / 28.42 | 27.41 / 27.26 | 27.88 / 28.15 |
| 4 pg / 4 fg | 31.65 / 32.55 | 32.53 / 32.00 | 31.41 / 32.41 | 30.79 / 30.54 | 30.83 / 31.37 |
| 400 fg / 400 ag | 35.05 / 36.23 | 36.47 / 36.06 | 35.29 / 35.58 | 33.94 / 33.58 | 33.59 / 34.57 |
| 40 fg / 40 ag | 38.64 / 39.69 | 39.65 / 39.12 | 38.44 / 39.10 | 37.01 / 36.95 | 37.64 / 36.52 |

measurements and evaluate the sensitivity of molecular identification. In this study, we optimized methods to extract mold (filamentous fungal) DNA. We did lysis of clinical and environmental mold using enzyme and Qiagen purification kit. However, their purity was not good for quantitative analysis, and many variables among species and strains. So, we tried to look for the efficient lysis and complete removal of protein and PCR inhibitors, and then it has been tested in hundreds of mold strains. First, freeze-thawing and bead-beating methods were used for mechanical lysis, and different lysis and protein removal solutions were used for the conidia and hyphae (hyphae, Trizol solution and isopropanol; conidia, Epicentre's lysis buffer and protein precipitation solution). Some researchers suggested bead beating could be more efficient at lysing fungus than enzymatic lysis [21]. And, Trizol reagent (Invitrogen) led to a better yield in the hyphae, but affected the purity in conidia as the handling process during supernatant isolation and showed the incomplete lysis in conidia of some strain. Therefore, we performed lysis conidia using different lysis buffers. In the purification process, the most commonly used method was column chromatography for final purification, and high-purity fungal DNA was obtained in almost all strains. For accurate quantification, Quant-iT™ Pico-Green™ dsDNA Reagent (Invitrogen) was used, which has a lower error rate compared with conventional OD methods and offers advantages in quantitative analysis and sensitivity measurements.

To establish a multiplex real-time PCR method with high specificity and sensitivity, *benA* and *cyp51A* were used to select a primer pair and probe specific to the major *Aspergillus* sections, and *benA* has been used as an important molecular marker to analyze the phylogenic relationship between filamentous ascomycetes [5, 22]. The genetic analysis of *benA* is useful to distinguish cryptic species, which is not possible with ITS sequencing. Although there are differences between species, approximately 560 bp (bt2ab region: exons 3–6) can be amplified for nucleotide sequence analysis. However, in this study, the region was designed to be shorter (exons 5–6) to be more suitable for the multiplex probe-based real-time PCR method, which improved efficiency while maintaining specificity (S1 Fig, Table 1). While developing a probe-based method for rapid simultaneous detection, we also designed *benA* primers for further sequence analysis to achieve specificity toward each species after amplification/probe detection. This could specifically amplify certain *Aspergillus* sections through multiplex real-time PCR, reflecting high specificity without cross-amplification of other fungal species. Further, quantification analysis of each probe showed an extremely small error rate, and approximately $10^1$ genomic copies could be detected, indicating high sensitivity. These results show that the developed method have similar sensitivity to existing methods used for detection and quantification analysis [4, 19].

Regarding the use of standard DNA as a control for a developed method, gDNA usage for each *Aspergillus* strain is complex and requires much time; furthermore, many standard DNA types for specific probes are needed. Therefore, in this study, synthetic DNA was designed and used as standard DNA that reacted with all probes. The developed *Aspergillus* positive control DNA was amplified in all *benA* probes that identified different *Aspergillus* sections. Further, reconstructed *Aspergillus* positive control DNA for quantitative analysis was used to compare analyses with a standard curve, which correlated with the standard gDNA curve (Table 5). These results indicate that this DNA can be used as the control for sample gDNA quantification, to verify the primers and probe used in the experiment, and to measure experimental sensitivity and specificity. Also, use of multiple control DNAs complicates the experiment and can increase the cost of diagnostics by using multiple wells. To solve this problem, control DNA was synthesized and used. The synthetic DNA could be applied with the newly designed *benA* probe in the field for IA diagnosis.

Among the sections amplified in the first stage, azole resistance in *A. fumigatus* and *A. tubingensis* of section *Nigri* can be detected using multiplex real-time PCR primer/probes designed in the second stage. Using isolates from *Aspergillus* section *Fumigati*, the melting peak and temperature were identified for the primers that specifically amplified the *cyp51A* TR region in *A. fumigatus* and gDNA subjected to a 10-fold serial dilution. Wild-type isolates and mutant isolates with TR34 showed different melting temperatures. In particular, *A. tubingensis* revealed intrinsically higher azole minimum inhibitory concentrations according to previous studies [2, 3, 5]. The *A. tubingensis*-specific primers and probe designed in this study showed that only *A. tubingensis* was amplified without affecting other *Aspergillus* in section *Nigri*.

Interestingly, we tried to use DNA extracted from early-stage fungal colonies for PCR analysis and determined that the minimum diameter and culture time required for successful PCR detection were 0.4 cm and 24 h, respectively. The same results were obtained for four probes, suggesting that fungal colony hyphae cultured for 24 h can be used for earlier molecular identification. Although there may be some limitations in early diagnosis, our developed method can dramatically reduce the minimum culture period for molecular identification within 2–3 days. In other words, it has been reduced by 3 to 14 days compared to the previous method.

In conclusion, first, newly designed primers and probes were used to conduct a multiplex real-time PCR assay, and high sensitivity and specificity was confirmed through clinical and environmental fungal isolates. Second, azole-resistant *A. fumigatus* and *A. tubingensis* were detected, and major *Aspergillus* sections were identified. Third, this real-time PCR method resulted in successful amplification specific to each section with DNA extracted from *Aspergillus* hyphae in the early phases of the cultured colony. This method can provide a template for rapid and quantitative diagnosis of IA. Fourth, *Aspergillus* positive control DNA designed for the standard control permitted quantitative analysis. It is expected that further research on new biomarkers will be conducted using the database obtained through this study. And we will approach the LOQ and LOD both in fungal DNA extraction and multiplex real time PCR from clinical specimens for clinical use in further study.

## Supporting information

**S1 Fig. Probe and primer design by sequence comparison analysis of major *Aspergillus* species.** The MegAlign Pro program from DNAStar Lasergene (version 15 software package) was used to compare the sequence analysis by using Clustal Omega alignment. The comparison of *benA* sequences in *Aspergillus* species. All *benA* sequences in *Aspergillus* were compared and regions specific to the section without affecting other sections were selected as probe candidates, and primers that included all selected regions were chosen. Finally, one pair of primers and five probes were selected.
(TIF)

**S2 Fig. Analysis of melting peak for two types of *Aspergillus fumigatus* showing different azole resistance.** Wild-type (WT) isolate without azole resistance in *A. fumigatus* and azole-resistant isolate with TR34 sequence were used in the experiment. A pair of primers that amplified the region in the *cyp51A* promotor known to be involved in azole resistance was used for the melting peak analysis. DNA was subjected to a 10-fold dilution (4 ng to 40 fg). The melting temperatures from the melting curve analysis were different: 83.0ºC ± 0.3ºC in WT and 85.6ºC ± 0.6ºC in azole-resistant type (n = 3).
(TIF)

**S3 Fig. SPUD analysis of SPUD plasmid DNA and genomic DNA.** Quantitative PCR was performed using positive control SPUD plasmid DNA ($1.3 \times 10^5$ copies / μL, n = 20) and

various genomic DNAs containing the same amount of SPUD (n = 70), and the results were plotted on a box plot. It can be seen that the Cq value between the two DNAs varies within 1 cycle. The experiment repeated three times.
(JPG)

**S1 Table. Non-Aspergillus species used for negative control in qualitative analysis.**
(DOCX)

## Author Contributions

**Conceptualization:** Sung-Yeon Cho, Dong-Gun Lee.

**Data curation:** Won-Bok Kim, Chulmin Park, Hye-Sun Chun.

**Investigation:** Won-Bok Kim, Chulmin Park, Hye-Sun Chun.

**Methodology:** Won-Bok Kim, Chulmin Park.

**Project administration:** Dong-Gun Lee.

**Supervision:** Dong-Gun Lee.

**Validation:** Won-Bok Kim, Chulmin Park, Sung-Yeon Cho.

**Writing – original draft:** Won-Bok Kim, Chulmin Park.

**Writing – review & editing:** Won-Bok Kim, Chulmin Park, Sung-Yeon Cho, Dong-Gun Lee.

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
