## [Decision Letter · Decision Letter 0]

30 Dec 2019

PONE-D-19-32117

Development of Multiplex Real-time PCR for Rapid Identification and Quantitative Analysis of Aspergillus Species

PLOS ONE

Dear Dr Lee,

Thank you for submitting your manuscript to PLOS ONE. After careful consideration, we feel that it has merit but does not fully meet PLOS ONE’s publication criteria as it currently stands. Therefore, we invite you to submit a revised version of the manuscript that addresses the points raised during the review process.

We would appreciate receiving your revised manuscript by Feb 10 2020 11:59PM. To enhance the reproducibility of your results, we recommend that if applicable you deposit your laboratory protocols in protocols.io, where a protocol can be assigned its own identifier (DOI) such that it can be cited independently in the future. For instructions see: http://journals.plos.org/plosone/s/submission-guidelines#loc-laboratory-protocols

We look forward to receiving your revised manuscript.

Kind regards,

Ruslan Kalendar, PhD

Academic Editor

PLOS ONE

"This research was supported by an industry-university research grant (5-2019-D0166-00004) through the Yuhan Corporation."

We note that one or more of the authors have an affiliation to the commercial funders of this research study : [Yuhan Corporation.].

2) Please also provide an updated Competing Interests Statement declaring this commercial affiliation along with any other relevant declarations relating to employment, consultancy, patents, products in development, or marketed products, etc.  

Reviewers' comments:

Reviewer's Responses to Questions

**Comments to the Author**

1. Is the manuscript technically sound, and do the data support the conclusions?

Reviewer #1: Yes

Reviewer #2: Yes

Reviewer #3: Yes

2. Has the statistical analysis been performed appropriately and rigorously? 

Reviewer #1: I Don't Know

Reviewer #2: N/A

Reviewer #3: I Don't Know

3. Have the authors made all data underlying the findings in their manuscript fully available?

Reviewer #1: Yes

Reviewer #2: Yes

Reviewer #3: Yes

4. Is the manuscript presented in an intelligible fashion and written in standard English?

Reviewer #1: Yes

Reviewer #2: Yes

Reviewer #3: No

5. Review Comments to the Author

Reviewer #1:

1. What are the novelties of your work compared to other studies using "multiplex real time PCR" for Aspergillus detection like: "Multiplex real-time PCR for detection and quantification of mycotoxigenic Aspergillus, Penicillium and Fusarium" 2009. Journal of Stored Products Research.

2. Lines 56-58, Introduction part: -These steps are shared between your method and "identification via the polymerase chain reaction" methods. - What are the merits of your method compared to other molecular methods such as multiplex PCR?

3. Line 215, discussion part: In average, how much time was reduced compared to other methods?

Reviewer #2:

The manuscript is relevant for PlosOne but there are some issues. You have a very good results but I would like that you explain more about material and methods used to synthesize the control DNA. When you explain about the standard curve, never show the efficiency (E). However in M&M is explianed that it has been calcualted.

Reviewer #3:

1-DNA extraction and purification

Authors indicated: “ In this study, we optimized methods to extract mold DNA. First, freeze-thawing and bead-beating methods were used for mechanical lysis, and different lysis and protein removal solutions were used for the conidia and hyphae (hyphae, Trizol solution and isopropanol; conidia, Epicentre’s lysis buffer and protein precipitation solution). Different lysis buffers were used because Trizol reagent”.

Please, for this issue referr to previous validations of fungal DNA extraction, where it was shown that columns and magnetic beads allowed collecting DNA and separate PCR inhibitors, but detection rates could not be related to DNA-avidity of beads or to elution but to the lack of proteolysis.

Please refer to:

-Goldschmidt P, Degorge S, Merabet L, Chaumeil C. Enzymatic treatment of specimens before DNA extraction directly influences molecular detection of infectious agents. PLoS One. 2014 Jun 17;9(6):e94886. doi:10.1371/journal.pone.0094886.)

-Hsu M, Chen K, Lo H, Chen Y, Liao M, et al. (2003) Species identification of medically important fungi by use of realtime LightCycler PCR. J. Med. Microbiol 52: 1071–1076.

2- As for potential clinical use it should be stressed that the yields of DNA extraction and the PCR inhibitors must be monitored (i.e. by adding internal controls to each sample). results of the systematic validation of this issue were not found in the manuscript.

3- Resutls of LOQ and LOD were 40 fg and 400 fg, respectively. Here, in addition to the quantitative analysis expressed as DNA mass (pg, fg, etc), the sensitivity of the test for each sample should be also related to the number of colony forming units.

For further clinical use, the detection limits should be be also compared with fresh titrated fungal suspensions seeded into the haemoculture system (inoculums containing serial diluted CFU/bottle).

Please, refer to:

-Obara H, Aikawa N, Hasegawa N, Hori S, Ikeda Y, et al. (2011) The role of a real-time PCR technology for rapid detection and identification of bacterial and fungal pathogens in whole-blood samples. J Infect Chemother 3: 327–33.

-Goldschmidt P, Degorge S, Che Sarria P, Benallaoua D, Semoun O, Borderie V, Laroche L, Chaumeil C. New strategy for rapid diagnosis and characterization of fungal infections: the example of corneal scrapings. PLoS One. 2012;7(7):e37660.)

4- Results of specificity of the different set of primers with no filamentous fungi were not found.

-Marr KA, Carter R, Crippa F, Wald A, Corey L (2002) Epidemiology and outcome of mould infections in hematopoietic stem cell transplant recipients. Clin Infect Dis 34: 909–17.

-Horvath L, George B, Murray C, Harrison L, Hospenthal D (2004) Direct comparison of the BACTEC 9240 and BacT/ALERT 3D automated blood culture systems for Candida growth detection. J Clin Microbiol 42: 115–8.

6. PLOS authors have the option to publish the peer review history of their article (what does this mean?). If published, this will include your full peer review and any attached files.

Reviewer #1: No

Reviewer #2: No

Reviewer #3: No

---

## [Author Response · Author response to Decision Letter 0]

8 Feb 2020

Answers to Editor’s & Reviewers’ comments 

- We made further edits in response to the editor's comments. The revised version was corrected by the style of the journal, and modified parts were given in highlight.

"This research was supported by an industry-university research grant (5-2019-D0166-00004) through the Yuhan Corporation."

We note that one or more of the authors have an affiliation to the commercial funders of this research study : [Yuhan Corporation.].

 2) Please also provide an updated Competing Interests Statement declaring this commercial affiliation along with any other relevant declarations relating to employment, consultancy, patents, products in development, or marketed products, etc. 

- As you indicated, we revised “Competing Interest” and attached it to the rebuttal letter. 

 

Review Comments to the Author

Reviewer #1:

 1. What are the novelties of your work compared to other studies using "multiplex real time PCR" for Aspergillus detection like: "Multiplex real-time PCR for detection and quantification of mycotoxigenic Aspergillus, Penicillium and Fusarium" 2009. Journal of Stored Products Research.

: I appreciate that you reviewed our paper in detail. In this study, our strategy for the molecular detection of Aspergillus focused on the diagnostics of the invasive aspergillosis (IA). Due to the high morbidity and mortality of IA, the rapid detection and identification are very important in the clinical setting. Aspergillus fumigatus, A. niger, A. flavus, and A. terreus have been reported as the significant species isolated in IA. Our designed probe can be specifically and sensitively to the major species of IA (A. fumigatus, A. niger, A. flavus, and A. terreus), in addition to the filamentous ascomycetes. Also, our work additionally detects to azole-resistant A. fumigatus and A. tubingensis of which strains have high MIC to azole comparing to other Aspergillus species. Taken together, our work could be used as the rapid identification and azole resistance in the diagnostic of IA, compared to other studies. In Discussion part, we discussed it in the revised version (line 297-302 ).

2. Lines 56-58, Introduction part: -These steps are shared between your method and "identification via the polymerase chain reaction" methods. - What are the merits of your method compared to other molecular methods such as multiplex PCR?

 : As you commented, “molecular identification is limited” should be unclear. We intended to improve the limitations of current time-consuming filamentous fungal identification by sequencing-based methods. Therefore, it is modified as “sequencing-based identification is time-consuming” (line 54-55).

Compared to multiplex PCR, our method has the advantages of the probe-based real time PCR, so it can be more sensitive, more specific, faster, and used in a quantitative method. Also, our probe-based method could be a time effective way because it does not need additional time for the sequencing (line 233-235 in the section Discussion).

3. Line 215, discussion part: In average, how much time was reduced compared to other methods?

: It usually takes 3-14 days of fungal culture for conidia production (Different culture time for each clinical strains), 2-3 days for PCR & sequencing, and 2-3 days for azole-resistant MIC analysis. Therefore, on average, it takes 7 to 20 days. However, in this study, we increased sensitivity and detected in the early stages of hyphae generation, and we performed probe-based real time PCR that did not perform sequencing. As a result, the time required for diagnosis after culture was reduced to 2 ~ 3 days. In other words, it has been reduced by 3 to 14 days compared to the previous method.

This has been modified in the introduction (line 54-57) and discussion (line 294-296).

 

Reviewer #2:

The manuscript is relevant for PlosOne but there are some issues. You have a very good results but I would like that you explain more about material and methods used to synthesize the control DNA. 

When you explain about the standard curve, never show the efficiency (E). However in M&M is explianed that it has been calcualted.

Line 127: What is the protocol used to synthesized the control DNA?

: I appreciate that you reviewed our paper in detail.

As you commented, our explanation could be insufficient. So we described more in materials and methods (lines 124-131) as below.

At first, we aligned each consensus sequence of benA in Aspergillus species (S1 Fig). We used benA sequence of A. fumigatus as the template and designed to have each probe sequence (conserved probe of filamentous ascomycetes, Fumigati, Nigri, Flavi, and Terrei) (Table 2). And then, the designed DNA was synthesized genetically (Bionics Corporation, Seoul). We cloned it into the pUC57 plasmid, and it was transformed into Escherichia coli BL21strain. Plasmid DNA of transformants was extracted, tested for the multiplex, and used as control DNA.

Line 130: why has the author synthesized a control DNA instead to clon different fragments of DNA as positives controls?

: If different fragments are used as controls, several control-set should be needed. More controls need more wells. In clinical settings, more control leads to fewer test samples, especially in quantitative analysis. Therefore we designed a synthesized control including all probes (5 probes). We describe it at the section of Material & Method and Discussion (line 124-131, line 279-281).

Results: Line 167: delete a "were not"

: I thank you for reviewing in detail. We had been corrected it (line 177-178). 

Line 188: The author should include the Efficiency (%) calculated, as explianed in M&M, to each standard curve.

: As you commented, we added efficiency (%) in the revised version. The amplification efficiency of each probe was found to be 89.8% ~ 104.8% in gDNA and 90.0% ~ 99.4% in Control DNA. These results are described in the line 192-193 of the results section, and each value is added in Table 5.

 

Reviewer #3:

1-DNA extraction and purification

Authors indicated: “ In this study, we optimized methods to extract mold DNA. First, freeze-thawing and bead-beating methods were used for mechanical lysis, and different lysis and protein removal solutions were used for the conidia and hyphae (hyphae, Trizol solution and isopropanol; conidia, Epicentre’s lysis buffer and protein precipitation solution). Different lysis buffers were used because Trizol reagent”.

Please, for this issue referr to previous validations of fungal DNA extraction, where it was shown that columns and magnetic beads allowed collecting DNA and separate PCR inhibitors, but detection rates could not be related to DNA-avidity of beads or to elution but to the lack of proteolysis.

Please refer to:

-Goldschmidt P, Degorge S, Merabet L, Chaumeil C. Enzymatic treatment of specimens before DNA extraction directly influences molecular detection of infectious agents. PLoS One. 2014 Jun 17;9(6):e94886. doi:10.1371/journal.pone.0094886.)

-Hsu M, Chen K, Lo H, Chen Y, Liao M, et al. (2003) Species identification of medically important fungi by use of realtime LightCycler PCR. J. Med. Microbiol 52: 1071–1076.

: I appreciate that you reviewed our paper in detail.

We reviewed the two papers which you referred to. Both papers showed better removal of PCR inhibitors using lyticase and protease K. In these papers you mentioned, only a few strains of filamentous fungus (mold) were tested. We did lysis of clinical and environmental mold using enzyme and Qiagen purification kit. However, their purity was not good for quantitative analysis, and many variables among species and strains. So, we tried to look for the efficient lysis and complete removal of protein and PCR inhibitors, and then it has been tested in hundreds of mold strains.

We used a bead beating lysis method using Trizol reagent (invitrogen) in the case of hyphae and lysis solution of MasterPure ™ Yeast DNA Purification Kit (Epicentre) in conidias. In addition, we performed purification using phenol/chloroform or precipitation buffer (kit), and then they were additionally purified by the column kit (Qiagen). In tested mold strains, these methods performed complete physical lysis and a fungal DNA purification with the high purity for the quantitative analysis. Trizol agent is more economical and effective in hyphae, but it showed the incomplete lysis in conidia of some strain. The lysis solution of the kit showed the complete lysis and removal of PCR inhibitor in both hyphae and conidia. 

Some researchers suggested bead beating could be more efficient at lysing fungus than enzymatic lysis (EAPCRI method for Aspergillus DNA from whole blood) (Rosemary A Barnes, P Lewis White, C Oliver Morton, Thomas R Rogers, Mario Cruciani, Juergen Loeffler, J Peter Donnelly, Diagnosis of aspergillosis by PCR: Clinical considerations and technical tips. Medical Mycology 2018 56(suppl_1):60-72. https://doi.org/10.1093/mmy/myx091).

We discussed it in the Discussion part (line 241-257)

2- As for potential clinical use it should be stressed that the yields of DNA extraction and the PCR inhibitors must be monitored (i.e. by adding internal controls to each sample). results of the systematic validation of this issue were not found in the manuscript.

: Actually, we monitored of PCR inhibitors using SPUD plasmid DNA (insert in pUC57 vector) as an internal control. It is a routine task and I did not write it in the paper. However, I regret that it was not mentioned in the first version of our paper. We add the monitoring results in the revised version.

We performed an experiment to determine the presence of a PCR inhibitor. In the real time PCR process, the experiment was conducted by spiking SPUD DNA with a 1.3 X 105 copy number. When the difference of Cq value was <1 compared to the SPUD DNA diluted in PBS, it was determined that there was no PCR inhibitor. Also, we checked the yields of DNA extraction by monitoring the mass of genomic DNA. When fungal DNA was extracted in ~108 of conidia, and the amount is >3.5μg (usually 4-5 μg) measured by Quant-iT™ PicoGreen™ dsDNA Reagent (Invitrogen), we used it in our study (the mass of one genome copy of Aspergillus was calculated to around 40 fg). We described it in the section of Material and Method and Result. (S3 Fig, line 159-166, 226-228).

3- Resutls of LOQ and LOD were 40 fg and 400 fg, respectively. Here, in addition to the quantitative analysis expressed as DNA mass (pg, fg, etc), the sensitivity of the test for each sample should be also related to the number of colony forming units.

For further clinical use, the detection limits should be be also compared with fresh titrated fungal suspensions seeded into the haemoculture system (inoculums containing serial diluted CFU/bottle).

Please, refer to:

-Obara H, Aikawa N, Hasegawa N, Hori S, Ikeda Y, et al. (2011) The role of a real-time PCR technology for rapid detection and identification of bacterial and fungal pathogens in whole-blood samples. J Infect Chemother 3: 327–33.

-Goldschmidt P, Degorge S, Che Sarria P, Benallaoua D, Semoun O, Borderie V, Laroche L, Chaumeil C. New strategy for rapid diagnosis and characterization of fungal infections: the example of corneal scrapings. PLoS One. 2012;7(7):e37660.)

: As you mentioned, it could be a better way to describe the sensitivity related to the number of colony-forming units (CFU). However, CFU of the filamentous fungus (mold) is difficult to be counted due to the existence of hyphae and conidia (live/dead), the different growth rates of each conidia, error rate of hemocytometer counting, and so on. Therefore, the quantitative analysis from CFU of mold is very variable, and the validation should be difficult. We decided to validate our method by DNA mass related to fungal genome copy. Although the graphs and graphs are the mass of DNA (pg, fg), the genome copies number was obtained using the genome size and molar concentration, and we described in the text (Fig 1, Line 193-197, 220-224).

Therefore, we statistically described multiplex real time PCR results of LOQ and LOD in this paper, not the method of fungal DNA extraction. As you mentioned, we will approach the LOQ and LOD both in fungal DNA extraction and multiplex real time PCR from clinical specimens for clinical use in further study.

We will discuss it in the section of Discussion (line 304-305).

4- Results of specificity of the different set of primers with no filamentous fungi were not found.

-Marr KA, Carter R, Crippa F, Wald A, Corey L (2002) Epidemiology and outcome of mould infections in hematopoietic stem cell transplant recipients. Clin Infect Dis 34: 909–17.

-Horvath L, George B, Murray C, Harrison L, Hospenthal D (2004) Direct comparison of the BACTEC 9240 and BacT/ALERT 3D automated blood culture systems for Candida growth detection. J Clin Microbiol 42: 115–8.

: We designed a probe able to detect filamentous ascomycetes. Besides, non-ascomycetes and non-filamentous ascomycetes were used in the specificity of the test, and these results were described (Line 171-173). In one of non-filamentous ascomycetes (No. 39, Table S1), and 8 of non-ascomycetes molds (No. 40-47, Table S1), there was no cross-reaction in all set of probes. In the designed real time PCR in this study, the fungal DNA of filamentous ascomycetes (No. 1-38, Table S1) was amplified specifically. (Table 3, Table S1)

---

## [Editor Report · Decision Letter 1]

11 Feb 2020

Development of Multiplex Real-time PCR for Rapid Identification and Quantitative Analysis of Aspergillus Species

PONE-D-19-32117R1

Dear Dr. Lee,

We are pleased to inform you that your manuscript has been judged scientifically suitable for publication and will be formally accepted for publication once it complies with all outstanding technical requirements.

With kind regards,

Ruslan Kalendar, PhD

Academic Editor

PLOS ONE

---

## [Editor Report · Acceptance letter]

18 Feb 2020

PONE-D-19-32117R1 

Development of Multiplex Real-time PCR for Rapid Identification and Quantitative Analysis of *Aspergillus* Species 

Dear Dr. Lee:

I am pleased to inform you that your manuscript has been deemed suitable for publication in PLOS ONE. Congratulations! Your manuscript is now with our production department. 

With kind regards,

on behalf of

Dr. Ruslan Kalendar 

Academic Editor

PLOS ONE